# Methyljasmonate Elicitation Increases Terpenoid Indole Alkaloid Accumulation in *Rhazya stricta* Hairy Root Cultures

**DOI:** 10.3390/plants8120534

**Published:** 2019-11-22

**Authors:** Amir Akhgari, Into Laakso, Hannu Maaheimo, Young Hae Choi, Tuulikki Seppänen-Laakso, Kirsi-Marja Oksman-Caldentey, Heiko Rischer

**Affiliations:** 1VTT Technical Research Centre of Finland Ltd., P.O. Box 1000, VTT, Espoo 02044, Finland; hannu.maaheimo@vtt.fi (H.M.); Tuulikki.Seppanen-Laakso@vtt.fi (T.S.-L.); Kirsi-Marja.Oksman@vtt.fi (K.-M.O.-C.); 2Department of Biochemistry, University of Turku, Turku 20014, Finland; 3Division of Pharmaceutical Biosciences, Faculty of Pharmacy, P.O. Box 56, University of Helsinki, Helsinki, 00014, Finland; into.laakso@helsinki.fi; 4Natural Products Laboratory, Institute of Biology, Leiden University, Sylviusweg BE, Leiden 72, 2333, The Netherlands; y.h.choi@biology.leidenuniv.nl

**Keywords:** *Rhazya stricta*, hairy root cultures, terpenoid indole alkaloids, methyl jasmonate elicitation, chemical analyses, principal component analysis (PCA)

## Abstract

Methyl jasmonate is capable of initiating or improving the biosynthesis of secondary metabolites in plants and therefore has opened up a concept for the biosynthesis of valuable constituents. In this study, the effect of different doses of methyl jasmonate (MeJA) elicitation on the accumulation of terpenoid indole alkaloids (TIAs) in the hairy root cultures of the medicinal plant, *Rhazya stricta* throughout a time course (one-seven days) was investigated. Gas chromatography-mass spectrometry (GC-MS) analyses were carried out for targeted ten major non-polar alkaloids. Furthermore, overall alterations in metabolite contents in elicited and control cultures were investigated applying proton nuclear magnetic resonance (^1^H NMR) spectroscopy. Methyl jasmonate caused dosage- and time course-dependent significant rise in the accumulation of TIAs as determined by GC-MS. The contents of seven alkaloids including eburenine, quebrachamine, fluorocarpamine, pleiocarpamine, tubotaiwine, tetrahydroalstonine, and ajmalicine increased compared to non-elicited cultures. However, MeJA-elicitation did not induce the accumulation of vincanine, yohimbine (isomer II), and vallesiachotamine. Furthermore, principal component analysis (PCA) of ^1^H NMR metabolic profiles revealed a discrimination between elicited hairy roots and control cultures with significant increase in total vindoline-type alkaloid content and elevated levels of organic and amino acids. In addition, elicited and control samples had different sugar and fatty acid profiles, suggesting that MeJA also influences the primary metabolism of *R. stricta* hairy roots. It is evident that methyl jasmonate is applicable for elevating alkaloid accumulation in “hairy root” organ cultures of *R. strica.*

## 1. Introduction

*Rhazya stricta* is a member of the Apocynaceae family and broadly spread from the Middle East to the northwestern of the Indian sub-continent [1]. It is abundantly utilized in traditional medicine for the cure of several illness. *R. stricta* is a prolific source of terpenoid indole alkaloids (TIAs) and has been extensively investigated since the early 1960s [2]. Hitherto, the total number of isolated and structurally characterized indole alkaloids from *R. stricta* has risen to more than 100 alkaloids [1]. Because of its wide use in indigenous medicine, the plant has been the focus of a number of pharmacological investigations using in vitro and animal models. Research has disclosed anticancer activities of the crude alkaloid extracts [3] and isolated pure components, e.g., rhazinilam [4], sewarine, and vallesiachotamine [5].

The biosynthesis and accumulation of several pharmaceutically valuable TIAs, e.g., vinblastine or vincristine, typically occur at less than 1% dry weight [6] and is tightly controlled in spatiotemporal patterns and influenced by various variables including vegetative phase, environmental tensions, and nutriment availability states [7]. Chemical synthesis of many of the secondary metabolites, particularly alkaloids, is limited due to complex chemical structures and challenges in achieving the proper stereochemistry [8]. Biotechnological methods, e.g., tissue, cell and organ cultures, therefore constitute promising production platforms [9].

Plants have evolved specific tactics to encounter environmental changes/stresses to support their sessile life. They often synthesize defense-associated secondary metabolites in response to environmental perturbation. Initial attack signals that are sensed by the plant are known as elicitors [10]. Elicitors include abiotic, e.g., metal ions and inorganic compounds, and biotic compounds such as pathogen cell walls or flagella, insect-derived cues or molecules that are spread from the injured plant cell walls. The elicitor signal recognition triggers a signal transduction and generates secondary signals finally leading to the stimulation of regulatory proteins (transcription factors), which coordinate the expression of biosynthetic genes [11]. Jasmonic acid (JA) or its derivatives, e.g., methyl jasmonate (MeJA) and salicylic acid (SA) are considered major secondary signaling molecules [12]. Production of these phytohormones causes a wide range of metabolic, physiological, and anatomical responses to the external stimuli. Jasmonic acid is an essential molecule in controlling many aspects of plant growth, development, and defensive response including secondary metabolism. Induction of hairy roots, as organ cultures, by *Agrobacterium rhizogenes* affords a persuasive and stable approach to elevate the biosynthesis of secondary metabolites [13]. Elicitation emerges as an efficient strategy for the improvement of secondary metabolite production in plant hairy root cultures [14] which mimics the plant’s natural response to environmental stress by stimulating JA biosynthesis leading to induction of the expression of genes for secondary metabolism and defense. In *Catharanthus roseus*, the transcript levels of a number of key TIA pathway genes, including tryptophan decarboxylase (*tdc*), geraniol 8-oxidase (*g8o*), strictosidine synthase (*str*), and strictosidine β-D-glucosidase (*sd*), increase significantly after exposure to JA [15,16,17]. Stimulation of secondary metabolite production in hairy root cultures by elicitors have been investigated in several studies [18]. Furthermore, differential gene expression analysis of elicited cultures is an effective strategy in the discovery of putative candidate genes in plant secondary metabolism, particularly in unsequenced species, e.g., *C. roseus* [16] and *Taxus* spp. [19].

In the past few decades only a limited number of cell culture studies of *R. stricta* [20] or somatic hybrids of *Rauvolfia serpentina* × *R. stricta* [21,22] have been reported to explore their TIA biosynthesis potential. However, the elicitation of TIAs has not been described in *R. stricta* plants, cell suspensions, or organ cultures. Recently, we have established hairy roots, as an organ culture system, for *R. stricta* [23]. *R. stricta* alkaloids contain a broad spectrum of chemical structures and polarities, therefore, we initially developed various analytical methods to qualify and quantify alkaloids [23,24,25]. In our previous study, GC-MS was particularly used for the analyses of non-polar alkaloids [24]. The content of characteristic non-polar alkaloids revealed the presence of 20 TIAs. In particular, we studied the accumulation of 12 major alkaloids including vincanine, eburenine, quebrachamine, fluorocarpamine, pleiocarpamine, tubotaiwine, tetrahydroalstonine, ajmalicine, yohimbine isomers, vallesiachotamine, and rhazine in 20 *R. stricta* hairy root clones. The other eight alkaloids had small broad peaks, which overlapped with more intense peaks obscuring their quantification.

In the current study, we show that MeJA has significant effects on the accumulation of ten *R. stricta* alkaloids. Furthermore, to get a broader view on overall changes in the metabolite patterns between the control and MeJA-treated samples and to find characteristic metabolites responsible for the discrimination, ^1^H NMR coupled with multivariate data analysis, i.e., principal component analysis (PCA), was employed. In addition, this is the first study on elicitor application in hairy roots of *R. stricta*.

## 2. Materials and Methods

### 2.1. Hairy Root Induction

Wild type hairy roots were initiated through co-incubation of leaves with the wild-type strain of *A. rhizogenes* LBA 9402 according to our previous study [23]. *R. stricta* hairy roots have been maintained for over six years by subculturing them at four week intervals on solid medium and kept in darkness.

### 2.2. Elicitation

MeJA elicitation was carried out to test its effect on alkaloid production in *R. stricta* hairy roots. MeJA (Duchefa; Haarlem, The Netherlands) stock solution was made by dissolving it in 40% (v/v) ethanol to achieve a concentration of 25 mM and then filter sterilization (0.22 µm). Root tips (500 mg) of a randomly chosen wild type hairy root clone, initially grown on hormone-free modified Gamborg B5 medium [26] solid medium, were transferred into a 500 mL-Erlenmeyer-flask containing 200 mL liquid medium and were incubated at 24 ± 1 °C in darkness on a rotary shaker (110 rpm) for four weeks to attain a fresh hairy root culture for the next step. After four weeks, white root tips (100 mg; approximately 1 cm long) were transferred to 50 mL-Erlenmeyer-flasks containing 20 mL medium. The growth conditions remained the same, too.

Twenty-one-day-old hairy root cultures were supplemented with MeJA at the following final concentrations 50, 100, and 200 µM MeJA. For control cultures equal volumes (40 µL, 80 µL, 160 µL) of 40% ethanol was applied, respectively. The roots were incubated under the same conditions as mentioned above and collected in two day intervals (1, 3, 5, and 7 days) after elicitation. The roots were rinsed with sterile water, vacuum-filtered and gently blotted with filter paper to remove excess water before measuring the fresh weight. The samples were flash-frozen in liquid nitrogen, freeze-dried, and the dry weight was recorded. The experiments were carried out in biological triplicate.

### 2.3. Extraction of Alkaloids and GC-MS Analysis

Alkaloid extraction from hairy roots and medium and GC-MS (Agilent, CA, USA) analyses were carried out as described in our previous report [24]. The quantification of major ten alkaloids was accomplished by dividing the base peak intensities, obtained from total ion current (TIC) analyses, by the base peak abundance of the internal standard (2,4′-dipyridyl; Tokyo Kasei, Tokyo, Japan).

### 2.4. Extraction of Compounds and Metabolic Profiling by ^1^H NMR

Hairy roots, which had been exposed to MeJA (50 and 100 µM) for three days as well as control cultures, were lyophilized. Fifty milligrams of ground samples were transferred to 2-mL Eppendorf tubes and were extracted with 750 µL of KH_2_PO_4_ buffer (pH 6.0) in deuterium oxide (D_2_O) containing 0.1% (trimethylsilylpropionic acid-*d_4_* sodium salt, TSP): 750 µL CH_3_OH-*d_4_*. The mixture was vortexed heavily for 1 min, sonicated for 10 min, and then centrifuged at 10,000 rpm for 10 min and then the supernatant was transferred to a microtube. After another centrifugation step the supernatant (600 µL) was transferred to a 5-mm SampleJet NMR tube (Bruker Biospin, Karlsruhe, Germany). CH_3_OH-*d_4_* (99.90%) and D_2_O (99.80%) were purchased from Euriso-top (Saint Aubin, France) and KH_2_PO_4_ from Riedel-de Haën (Seelze, Germany). 

The NMR spectra were recorded at 22 °C on a 600 MHz Bruker Avance III NMR spectrometer (Bruker Biospin, Karlsruhe, Germany) equipped with a QCI cryoprobe (Bruker Biospin, Karlsruhe, Germany) and SampleJet (Bruker Biospin, Karlsruhe, Germany) automated sample changer. The residual water signal was suppressed by 4 s pre-saturation using the Bruker’s noesygppr1D pulse program. 128 scans with 16 dummy scans were accumulated from each sample. 64 k data points were recorded covering a spectral width of 20 ppm. Line broadening of 0.5 Hz was applied to the spectra prior to Fourier transformation. The ^1^H NMR spectra were automatically reduced to ASCII files.

### 2.5. Statistical Analysis

Statistical analyses were carried out using SPSS (version 21.0, Chicago, IL, USA). The experimental data for determining alkaloid levels in elicited and control hairy root cultures were subjected to Student’s *t*-test for pairwise comparison of means from two groups (control versus elicited sample). Data were expressed as mean ± RSD%. Principal component analysis (PCA) was employed with SIMCA-P + 13.0 (Umetrics, Umea, Sweden) software for ^1^H NMR data analyses based on Pareto scaling. Spectral intensities were scaled to total intensity of internal standard (TSP) at 0.0 ppm and reduced to integrated regions of equal width (0.04) using AMIX software (v.3.7 Bruker Biospin). The regions of *δ* 4.7–4.9 and δ 3.28–3.34 were excluded from the analysis because of the residual signal of the deuterated solvents (D_2_O and CH_3_OH-*d_4_*).

## 3. Results and Discussion

### 3.1. Effect of Elicitation on Alkaloid Accumulation

The GC-MS results revealed that all three concentration levels (50 µM, 100 µM, 200 µM) of MeJA statistically (student’s *t* test, *p* < 0.05) improved the accumulation of eburenine, fluorocarpamine, and quebrachamine at four different periods as compared with control cultures (Figure 1). The same tendency was found for pleiocarpamine, except for hairy roots that were exposed to 100 µM MeJA over a three-day period or treated with 200 µM MeJA for one day.

Five and seven days post-elicitation of hairy roots with 50 µM MeJA resulted in the maximum concentrations of eburenine, three-fold higher than the non-elicited cultures. The highest amount of quebrachamine was obtained on the first and fifth day after root cultures were treated with 50 µM MeJA and led to 2.6- and 2.5-fold increase compared with those of non-elicited cultures, respectively. In comparison to the control cultures, the content of fluorocarpamine was 3.7-fold increased on the fifth day after cultures were treated with 200 µM MeJA.

The maximum level of pleiocarpamine was 1.7- or 1.5-fold compared with the non-elicited cultures when experimental samples were treated with 50 µM MeJA for five or seven days, correspondingly. Tubotaiwine levels were significantly elevated in hairy roots, compared to control cultures, after three, five, and seven days of being exposed to 200 µM MeJA. Moreover, tubotaiwine levels were also increased after five and seven days of being exposed to 100 µM MeJA. The highest levels of tubotaiwine were obtained when hairy roots were treated with the maximum concentration of MeJA (200 µM) five or three days post-elicitation, with approximately a two-fold rise (1.9- and 1.7-fold, respectively) compared with their corresponding controls (Figure 1). Other experimental conditions did not alter the production of tubotaiwine.

The accumulation of tetrahydroalstonine had a significant increase in root cultures treated with the lowest concentration of MeJA (50 µM) three and five days post-elicitation and revealed 1.4- and 1.2-fold rises compared to control cultures (Figure 1). In addition, improved accumulation of tetrahydroalstonine was observed after hairy roots were treated with 100 µM and 200 µM for one and three days, respectively. Treatment of samples with various concentrations of MeJA seven days post-elicitation repressed the accumulation of tetrahydroalstonine; however, other experimental conditions did not statistically change its production compared with controls.

Only exposure of hairy roots to 50 µM MeJA on days one, three, and five significantly increased the accumulation of ajmalicine and resulted in 1.4-, 1.3-, and 1.5-fold rises, respectively, compared with non-elicited samples. Compared to controls, exposure of roots to highest concentration of MeJA (200 µM) at two different time spans (three and five days) and at concentrations between 50–200 µM MeJA for seven days did not influence the ajmalicine production. In other experimental conditions, the concentration even suppressed the ajmalicine accumulation. It was also observed that MeJA decreases vincanine, yohimbine (isomer II) and vallesiachotamine production (Figure 1). The chemical structures of all target alkaloids are presented in Figure 2. Total ion GC-MS and GC-MS spectra of the target alkaloids are illustrated in Appendix A, respectively.

GC-MS results revealed dose- and/or time-dependent increase of seven alkaloids and eburenine, quebrachamine, pleiocarpamine, and fluorocarpamine were the most responsive alkaloids to MeJA-elicitation. It should be pointed out that in our previous study on *R. stricta* alkaloids twelve major alkaloids, from twenty hairy root clones, were quantified by GC-MS analyses [23]. However, in the current investigation ten alkaloids were quantified by GC-MS. The two remaining alkaloids, rhazine and yohimbine isomer I, showed considerable peak broadening and partial overlapping with the neighboring peaks, therefore, their quantification remained unreliable and was not included in the analyses.

Monoterpene indole alkaloids are generated via rearrangement of the glycosylated central intermediate, strictosidine. In the first switching point deglucosylation is catalyzed by the substrate-specific strictosidine β-*D*-glucosidase to yield the highly reactive open-ring dialdehyde intermediate [27]. The unstable dialdehyde is converted to 4,21-dehydrocorynantheine aldehyde, followed by spontaneous conversions to yield precursors (cathenamine and 4,21 dehydrogeissoshizine) for additional alkaloid scaffolds.

Strictosidine rearrangement can yield cathenamine, biogenetically an important intermediate for the biosynthesis of corynanthe type alkaloids including ajmalicine and tetrahydroalsonine. Two different cathenamine reductases are known: one converting cathenamine to ajmalicine [28] and the other cathenamine to tetrahydroalstonine [29]. Recently, several medium chain dehydrogenases/reductases that produce the heteroyohimbine stereoisomers ajmalicine and/or tetrahydroalstonine were discovered, too [30]. We postulate that time- and dose-dependent MeJA-elicitation in *R. stricta* hairy roots possibly led to the overexpression of these genes and consequently resulted in higher accumulation of ajmalicine and tetrahydroalstonine.

One of the most important rearrangements of strictosidine is its conversion into 4,21 dehydrogeissoshizine and subsequently to the biosynthetic intermediate preakuammicine, the precursor for the strychnos and aspidosperma alkaloids. Eburenine and quebrachamine belong to aspidosperma- and vincanine to strychnos-type alkaloids [20,31]. Tubotaiwine is a member of the aspidospermatan alkaloids. The formation of the aspidospermatan alkaloids is biogenetically related to that of the strychnan alkaloids [32,33]. Fluorocarpamine and pleiocarpamine belong to ajmaline-sarpagine type alkaloids [32], generated in a number of steps from a dehydrogeissoschizine precursor via the early intermediate polyneuridine aldehyde [27]. Vallesiachotamine biosynthesis is initiated directly from deglycosylated strictosidine, a dialdehyde, which is not a common intermediate of TIA biosynthesis [34]. Yohimbine is a carbocyclic variant related to ajmalicine and is likely to arise from dehydrogeissoschizine by homoallylic isomerization of the keto dehydrogeissoschizine [35,36].

It appears that MeJA elicitation of *R. stricta* hairy roots channels the flux toward aspidosperma-, aspidospermatan- and ajmaline-sarpagine-type alkaloids while strychnos-type vincanine accumulation is either significantly suppressed or not changed.

The addition of ethanol in *Rhazya* control hairy root cultures could potentially influence the alkaloid accumulation. Gerasimenko et al. [37] reported that addition of ethanol in *R. serpentine* × *R. stricta* cell cultures lowered the content of major components. Contrarily, a pronounced enhancing elicitation effect of ethanol was described for tropane alkaloids in *Anisodus* hairy roots [38]. This could partly explain the differences between the average content of eburenine and vincanine in *R. stricta* wild type hairy roots analyzed in our previous study [24] and control cultures in the current experiments. However, it is important to stress that the elicitation experiments were performed years after the initial characterization and with different clones. Both long-term stability [39,40] and instability [41,42] have been described for hairy roots in terms of secondary metabolite production. In this study, a small but significant increase of ajmalicine content was observed in samples elicited with 50 µM MeJA, but not with higher concentration, for one, three, and five days in comparison to the control cultures. The same pattern was found in *Rhazya* hybrid cultures treated with 100 µM MeJA one or five days post-elicitation [21]. This was different from the results of Goklany et al. [43] who stated that MeJA-elicited *C. roseus* hairy root contained higher amounts of ajmalicine. This finding is also in contrast with reports indicating that ajmalicine accumulates higher in *C. roseus* cell suspensions elicited with 100 µM MeJA for six days and with 100 and 250 µM MeJA one to three days than in representative non-elicited samples [44,45]. It is worth noting that elicitation, mentioned in the above literature, was carried out in different growth stages than in *R. stricta* hairy roots. It might possibly explain the differences in the content of certain compounds after exposure to the same elicitor for a given time.

In the current experiments, TIAs were not detected in culture media. Nevertheless, Sheludko et al. [21] reported six indole alkaloids in media samples of *Rhazya* hybrid cultures. It appears here that elicitation of *R. stricta* hairy roots did not result in increased permeability of cell walls. Therefore, alkaloids are stored intracellularly and are not released into the medium. Sheludko et al. [21] showed that application of 100 µM MeJA one and five days post-elicitation elevated the concentration of several alkaloids including vallesiachotamione and yohimbine in *R. serpentina* × *R. stricta* hybrid cell culture. However, Sheludko et al. [22] showed that in another hybrid line the accumulation of eburenine, tubotaiwine, and vallesiachotamione was not induced after MeJA elicitation. Similarly, the elicitation of *R. stricta* hairy roots did not stimulate accumulation of vincanine, yohimbine (isomer II) and vallesiachotamine. On the contrary, tubotaiwine content was significantly higher in *Rhazya* hairy root cultures exposed to 100 µM MeJA for five days compared to control cultures and even approximately doubled in elicited samples with 200 µM MeJA for five days. Results of these studies evidently exhibited distinctive accumulation patterns of TIAs in the MeJA-elicited hairy roots of *R. stricta* and its cell cultures.

### 3.2. Metabolite Profile Changes after Elicitation Monitored by NMR

By using CH_3_OH-*d_4_*-KH_2_PO_4_ buffer in D_2_O (1:1, v/v), a wide range of primary (e.g., sugars) and secondary metabolites (indole alkaloids) could be monitored. The chemical shift (δ) of the ^1^H NMR spectra were segmented into three distinct regions: the region δ 6.00–δ 10.00 ppm, δ 3.00–δ 6.00 ppm, and δ 0.00–δ 3.00 ppm corresponds to aromatics, sugars, and organic and amino acids, respectively. Chemometric methods were applied to evaluate the overall alterations in the metabolic profiles between MeJA-elicited and control hairy roots of *R. stricta*. PCA was performed to the binned ^1^H NMR signals. As presented in Figure 3a, PCA distinctly discriminated between MeJA-elicited (50 and 100 µM) hairy roots and the control samples. Principal component 1 (PC1) clearly separated the samples into elicited and control samples, thereby explaining 85% of the variance. Controls tend to cluster together along the positive PC1 as well as positive PC2 (11%) scores whereas elicited samples with 50 and 100 µM MeJA both grouped on negative PC1 scores but clustered on the negative and positive PC2 scores, respectively (Figure 3a). Aromatic compounds (δ 6.00–δ 10.00 ppm) as well as organic and amino acids (δ 0.00–δ 3.00 ppm) were found at higher levels in elicited samples (Figure 3b). In addition, the compositions of sugars also were different between treated and control samples (Figure 3b). The current NMR multivariate data analysis also revealed different profiles of fatty acids among the elicited and non-elicited cultures.

The summation of ^1^H NMR spectral intensity from binned data ranging δ 0.49–δ 0.54 (ppm) was used to quantify total vindoline-type (related to the aspidosperma group) alkaloids in the hairy roots. It is evident that MeJA-treated hairy root cultures accumulated approximately five-fold higher total vindoline-type alkaloids compared to control cultures, whereas no statistically significant variation was found in hairy roots exposed to 50 and 100 µM MeJA (Figure 4).

Evidence showed that endogenous JA accumulation occurs following application of MeJA that subsequently leads to elevated accumulation of TIAs [12]. In addition, pathways in primary metabolism are induced by MeJA which in turn lead to the formation of TIA intermediates [7]. In *R. stricta* hairy roots, the elicited samples had higher contents of aliphatic compounds, organic acids, and amino acids than control cultures. These results are in accordance with the finding from jasmonic acid- [46,47] and salicylic acid-elicited [48] *C. roseus* cell suspension cultures monitored by NMR where the levels of several TIAs, some aliphatic amino acids and organic acids were significantly higher than in the control cultures.

Elicited and control cultures of *R. stricta* exhibited different carbohydrate profiles. Recently, Saiman et al. [47] found that sugar levels decreased by time in JA-elicited *C. roseus* cell suspension cultures after one day compared to the control. Similarly, it was found that glucose and sucrose reduced in MeJA-treated *Brassica rapa* leaves [49] and fungal-elicited *Papaver somniferum* cell cultures [50]. On the contrary, in *Cannabis sativa* cell suspension cultures JA-elicitation did not alter the sucrose and glucose levels [51]. It was postulated that upon jasmonate elicitation, sugars are catabolized to activate TCA cycle to generate energy (ATP) for the biosynthesis of defense-related compounds and their precursors [48,52].

In the current NMR experiment, CH_3_OH-*d_4_*-D_2_O extracts were used; therefore, the fatty acid and lipid signals were less prominent. However, the ^1^H NMR spectral region around 1.3 ppm represented the signals for fatty acids and lipids. Elicited *Rhazya* hairy roots had higher contents of fatty acids than controls (Figure 3b). This finding is in line with the result of *C. roseus* cell cultures exposed to JA for one-day prompting high accumulation of fatty acids [46]. The different profiles of fatty acids in the elicited and control cultures indicate the effect of MeJA on the metabolism of fatty acids which, in turn, induces accumulation of elicitor precursor and the accumulation of TIAs as delayed stress reactions to JA.

## 4. Conclusions

We describe here the elicitation of terpenoid indole alkaloids (TIAs) in *R. stricta* hairy roots. The study shows that MeJA, a well-known elicitor triggers TIA production in *Rhazya* hairy roots resulting in a five-fold rise in the accumulation of the total vindoline-type alkaloid in hairy roots, determined by NMR analyses. GC-MS analyses show dose and time-dependent increase in the amounts of seven out of ten targeted alkaloids in comparison with non-elicited control cultures. This study shows metabolic alterations upon MeJA elicitation in *R. stricta* and directing the carbon backbones to the biosynthesis of different precursors and terpenoid biosynthetic pathways. These results may provide an approach for developing metabolic engineering strategies of *R. stricta* in order to obtain high TIA-producing cultures.

## Figures and Tables

**Figure 1 plants-08-00534-f001:**
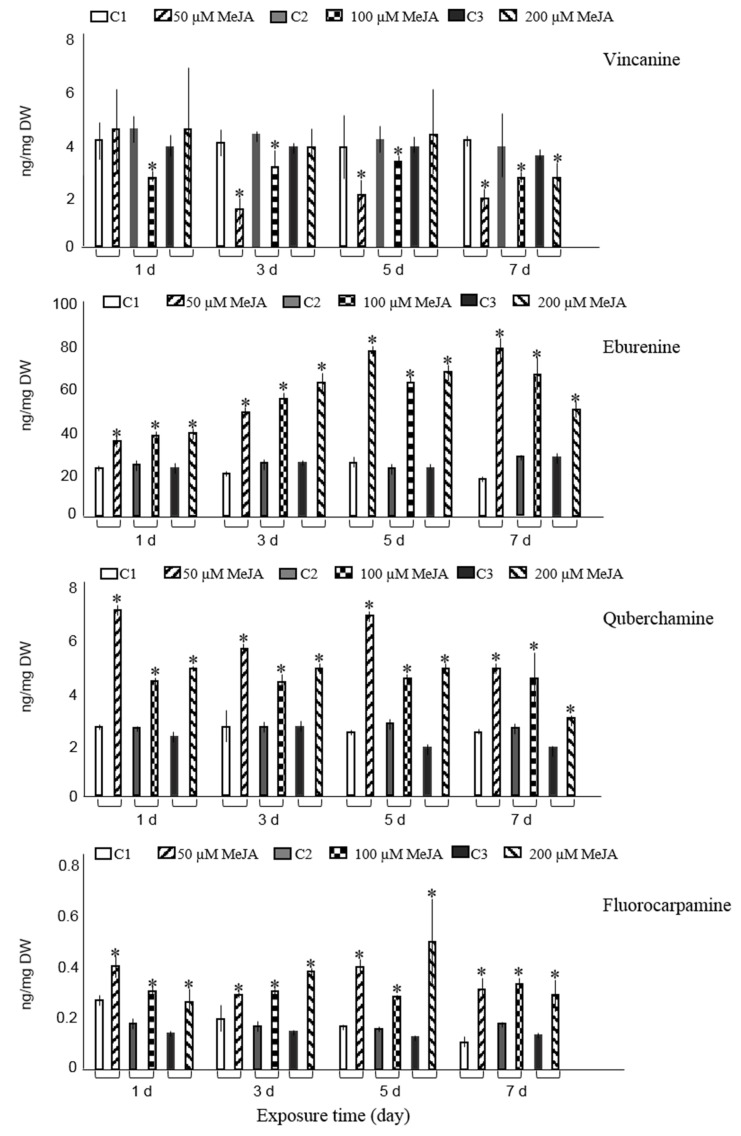
Accumulation of ten major alkaloids analyzed by GC-MS in *R. stricta* hairy roots elicited with 50, 100, and 200 µM MeJA after 1, 3, 5, and 7 days (d) exposure time. Control cultures (C1, C2, and C3) contained equal volume of 40% ethanol. Pairwise comparison of means from two groups (control versus elicited sample) is presented as mean ± SD of three samples for each treatment. * indicates significant difference in the mean between control and elicited sample (student’s *t* test, *p* < 0.05).

**Figure 2 plants-08-00534-f002:**
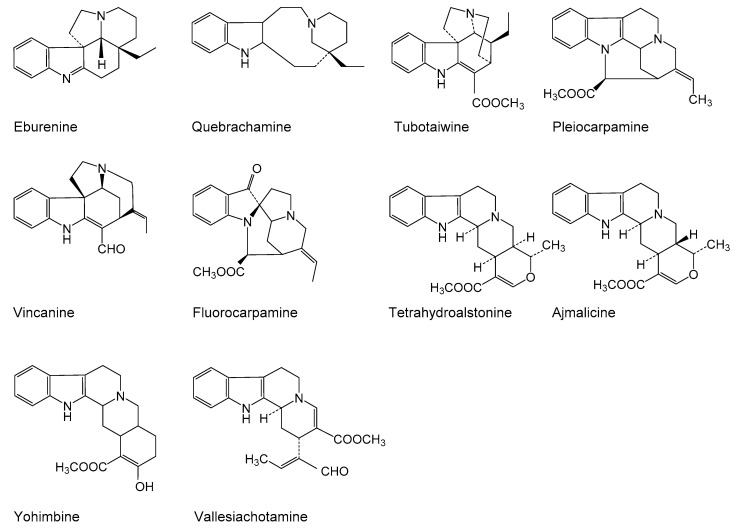
Structures of quantified alkaloids in the present study.

**Figure 3 plants-08-00534-f003:**
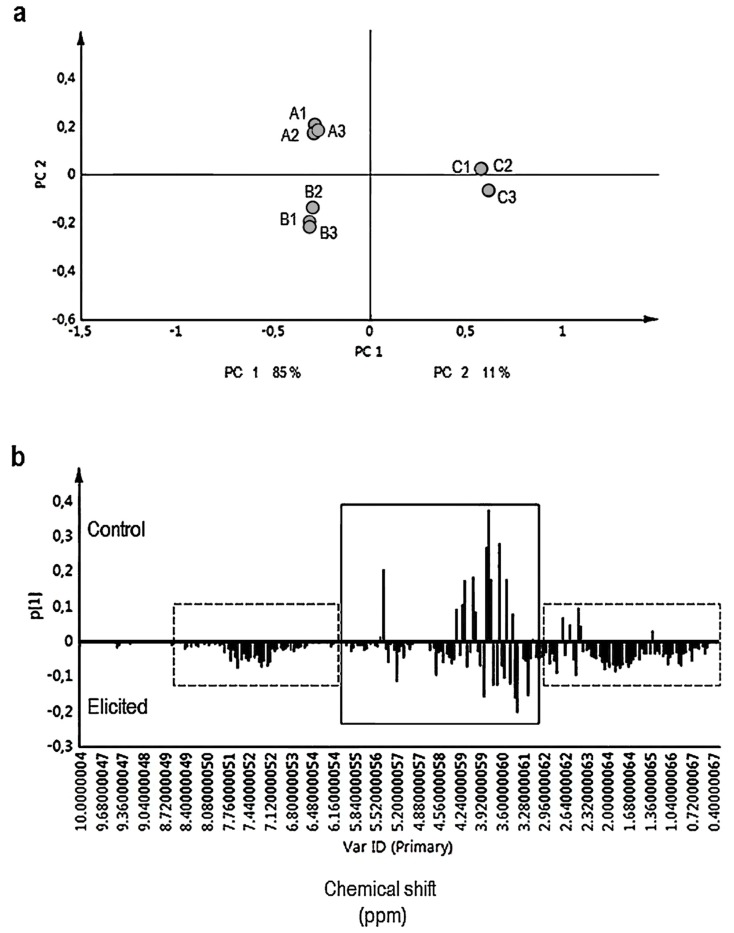
(**a**) Principal component analysis (PCA) score plot and (**b**) loading column plot based on ^1^H-NMR data from *R. stricta* hairy roots exposed to 50 μM (A1, A2, A3) and 100 μM (B1, B2, B3) MeJA for three days. In control cultures (C1, C2, and C3) equal volumes of 40% ethanol were applied (the same volume of 100 μM MeJA). All samples were in triplicate. Elicited samples were effectively separated by PC1 (**a**). The column plot (**b**) shows higher levels of aromatics (δ 6.00–δ 10.00 ppm) and organic/amino acids (δ 0.00–δ 3.00 ppm) (dashed lines) in elicited hairy root cultures than controls. Continuous line indicates different sugar and fatty acid profiles in elicited and control hairy root cultures.

**Figure 4 plants-08-00534-f004:**
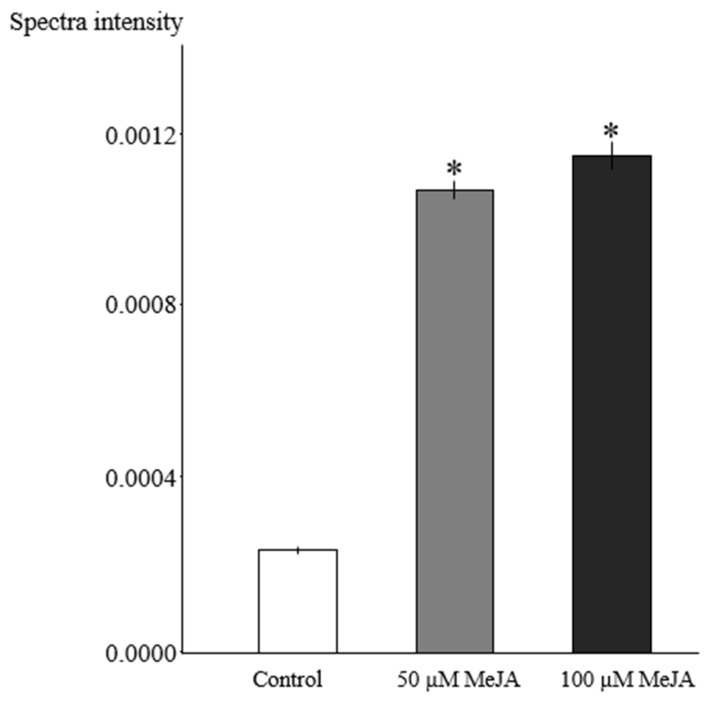
Comparison of total vindoline-type alkaloid based on ^1^H-NMR data in MeJA-elicited (50 and 100 μM, for three days) *R. stricta* hairy roots and control cultures (equal volumes of 40% ethanol were added as in 100 µM MeJA). The sum of the values of spectral intensity from ^1^H NMR bucket data in the region δ 0.49–δ 0.54 (ppm) was used for quantification of total vindoline-type alkaloids. Pairwise comparison of means from two groups (control versus elicited sample) is presented as mean ± SD of three samples for each treatment. * indicates significant difference in the mean between control and elicited sample (student’s *t* test, *p* < 0.05).

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
