# Peer review of "Methyljasmonate Elicitation Increases Terpenoid Indole Alkaloid Accumulation in Rhazya stricta Hairy Root Cultures"

_plants, 2019, doi:10.3390/plants8120534_

Round 1

Reviewer 1 Report

The manuscript describes the accumulation of terpenoid indole alkaloids (TIAs) as a result of MeJa treatment on R. stricta hairy root cultures. The document is relatively brief. However, it is well written, and the presented results (what little of them the authors deemed to show) are sufficiently discussed. The technical side of the work is also mostly correct, with the exemplary application of H-NMR spectroscopy to investigate changes in metabolic profiles due to elicitation.
Nevertheless, a few aspects of the work could be expanded and improved, which will presumably also bring the manuscript to the genuine "research article" size.
First, while the accumulation of TIAs upon elicitation is indeed noteworthy, the authors quite scantly explain why these particular alkaloids were accumulated. In other words, the presented levels of alkaloids do not translate into any insight on the actual biosynthetic pathways/enzymatic activities that are induced by MeJa.
Second, in addition to precise, quantitative GC-MS analyses of selected TIAs, a screening procedure should be applied to investigate changes in the levels of other alkaloids (perhaps on the % of control basis, if the appropriate standards are not available). Without any doubt, such an experiment would significantly increase the understanding of processes occurring upon the MeJa elicitation.
Third, just signaling changes in profiles of organic, amino, and fatty acids upon elicitation is, in my opinion, insufficient. Since the authors do have access to GC-MS, obtaining profiles of these compounds in control and elicited cultures should be relatively easy, and providing a more detailed description of the changes will definitely increase the value of this manuscript.
On the related note, the quality of the supplied figures is quite poor - they look somewhat hazy; maybe this is because of downsampling for the peer review copy, but the authors should look into this. Also, the contents of TIAs are in the less than ng to a few ng/mg DW range, but the axes of graphs are in μg/mg. Get rid of these zeros, please.

Author Response

Dear Gill Tan,

We appreciate the opportunity to revise our manuscript according to your and the reviewer’s valuable criticism. Please find a point-by-point list of answers addressing all requests below. Additionally, the English of the manuscript has been carefully edited (cf. track changes) to fulfill the high standards of “Plants”.

Response to the Reviewer 1:

Comments and Suggestions for Authors

Question 1:

The manuscript describes the accumulation of terpenoid indole alkaloids (TIAs) as a result of MeJa treatment on R. stricta hairy root cultures. The document is relatively brief. However, it is well written, and the presented results (what little of them the authors deemed to show) are sufficiently discussed. The technical side of the work is also mostly correct, with the exemplary application of H-NMR spectroscopy to investigate changes in metabolic profiles due to elicitation. Nevertheless, a few aspects of the work could be expanded and improved, which will presumably also bring the manuscript to the genuine "research article" size.

First, while the accumulation of TIAs upon elicitation is indeed noteworthy, the authors quite scantly explain why these particular alkaloids were accumulated. In other words, the presented levels of alkaloids do not translate into any insight on the actual biosynthetic pathways/enzymatic activities that are induced by MeJa.

Answer 1:

We appreciate the question and incorporated text (four paragraphs) in the manuscript to answer the question. Original lines 210-217 were replaced.

Replaced text (210-217):

“Vincanine belongs to strychnos-, eburenine and quebrachamine to aspidosperma-, and serpentine, tetrahydroalstonine, ajmalicine and yohimbine to corynanthe-type alkaloids [2,27]. Presumably, aspidosperma and strychnos alkaloids are derived from the corynanthe alkaloids [28].

Corynanthe-alkaloid yohimbine is derived from 4,21-dehydrogeissoshizine whereas ajmalicine, another member of corynanthe group, is generated from cathenamine [30]. Ajmalicine is further converted to serpentine and tetrahydroalstonine by two different reductases [31]. It seems that MeJA elicitation of R. stricta hairy roots channels the flux towards cathenamine and subsequently to the production of ajmalicine, tetrahydroalstonine and serpentine.”

New text (225-252, cf. track change in manuscript)

“Monoterpene indole alkaloids are generated via rearrangement of the glycosylated central intermediate, strictosidine. In the first switching point deglucosylation is catalyzed by the substrate-specific strictosidine β-D-glucosidase to yield the highly reactive open-ring dialdehyde intermediate [27]. The unstable dialdehyde is converted to 4,21-dehydrocorynantheine aldehyde, followed by spontaneous conversions to yield precursors (cathenamine and 4,21 dehydrogeissoshizine) for additional alkaloid scaffolds.

Strictosidine rearrangement can yield cathenamine, biogenetically an important intermediate for the biosynthesis of corynanthe type alkaloids including ajmalicine and tetrahydroalsonine. Two different cathenamine reductases are known: one converting cathenamine to ajmalicine [28] and the other cathenamine to tetrahydroalstonine [29]. Recently, several medium chain dehydrogenases/ reductases that produce the heteroyohimbine stereoisomers ajmalicine and/or tetrahydroalstonine were discovered, too [30]. We postulate that time- and dose-dependent MeJA-elicitation in R. stricta hairy roots possibly led to the overexpression of these genes and consequently resulted in higher accumulation of ajmalicine and tetrahydroalstonine.

One of the most important rearrangements of strictosidine is its conversion into 4,21 dehydrogeissoshizine and subsequently to the biosynthetic intermediate preakuammicine, the precursor for the strychnos and aspidosperma alkaloids. Eburenine and quebrachamine belong to aspidosperma- and vincanine to strychnos-type alkaloids [20,31]. Tubotaiwine is a member of the aspidospermatan alkaloids. The formation of the aspidospermatan alkaloids is biogenetically related to that of the strychnan alkaloids [32,33]. Fluorocarpamine and pleiocarpamine belong to ajmaline-sarpagine type alkaloids [32], generated in a number of steps from a dehydrogeissoschizine precursor via the early intermediate polyneuridine aldehyde [27]. Vallesiachotamine biosynthesis is initiated directly from deglycosylated strictosidine, a dialdehyde, which is not a common intermediate of TIA biosynthesis [34]. Yohimbine is a carbocyclic variant related to ajmalicine and is likely to arise from dehydrogeissoschizine by homoallylic isomerization of the keto dehydrogeissoschizine [35,36].

It appears that MeJA elicitation of R. stricta hairy roots channels the flux towards aspidosperma-, aspidospermatan- and ajmaline-sarpagine-type alkaloids while strychnos-type vincanine accumulation is either significantly suppressed or not changed.”

Updated References

O’Connor, S. E.; Maresh, J. J. Chemistry and biology of monoterpene indole alkaloid biosynthesis. Nat. Prod. Rep. 2006, 23, 532–547. Stöckigt, J. Indirect involvement of geissoschizine in the biosynthesis of ajmalicine and related alkaloids. J. Chem. Soc. Chem. Commun. 1978, 1097–1099. Stavrinides, A.; Tatsis, E. C.; Foureau, E.; Caputi, L.; Kellner, F.; Courdavault, V.; O’Connor, S. E. Unlocking the diversity of alkaloids in Catharanthus roseus: Nuclear localization suggests metabolic channeling in secondary metabolism. Chem. Biol. 2015, 22, 336–341. Stavrinides, A.; Tatsis, E. C.; Caputi, L.; Foureau, E.; Stevenson, C. E. M.; Lawson, D. M.; Courdavault, V.; O’Connor, S. E. Structural investigation of heteroyohimbine alkaloid synthesis reveals active site elements that control stereoselectivity. Nat. Commun. 2016, 7. Buckingham, J.; Baggaley, K. H.; Roberts, A. D.; Szabó, L.F. Dictionary of Alkaloids, 2nd ed.; Eds.; CRC Press/Taylor & Francis Group: London, UK, 2010. Saxton, J. E. Recent progress in the chemistry of the monoterpenoid lndole alkaloids. Nat. Prod. Rep. 1995, 4, 385-411. Szabó, L. F. Rigorous biogenetic network for a group of indole alkaloids derived from strictosidine. Molecules 2008, 13, 1875–1896. Shen, Z.; Eisenreich, W.; Kutchan, T. M. Bacterial biotransformation of 3α(S)-strictosidine to the monoterpenoid indole alkaloid vallesiachotamine. Phytochemistry 1998, 48, 293–296. Kan-Fan, C.; Husson, H. P. Biomimetic synthesis of yohimbine and heteroyohimbine alkaloids from 4,21-dehydrogeissoschizine. Tetrahedron Lett. 1980, 21, 1463–1466. Dewick, P.M. Medicinal Natural Products: A Biosynthetic Approach, 2nd ed.; Wiley: New York, NY, USA, 2002; pp. 350–358.

Question 2:

Second, in addition to precise, quantitative GC-MS analyses of selected TIAs, a screening procedure should be applied to investigate changes in the levels of other alkaloids (perhaps on the % of control basis, if the appropriate standards are not available). Without any doubt, such an experiment would significantly increase the understanding of processes occurring upon the MeJa elicitation.

Answer 2:

The applied GC-MS method indeed allows covering a wide spectrum of alkaloids (20 compounds). However, exact quantitative monitoring is only reliable for 12 major alkaloids as described in lines 88-92 and 204-209 and we argue that it wouldn’t be good scientific practice to present even relative ratios of the remaining alkaloids.

Lines 86-92: “In our previous study, GC-MS was particularly used for analyses of non-polar alkaloids [24]. The content of characteristic non-polar alkaloids revealed the presence of 20 TIAs. In particular, we studied the accumulation of 12 major alkaloids including vincanine, eburenine, quebrachamine, fluorocarpamine, pleiocarpamine, tubotaiwine, tetrahydroalstonine, ajmalicine, yohimbine isomers, vallesiachotamine and rhazine in 20 R. stricta hairy root clones. The other eight alkaloids had small broad peaks, which overlapped with more intense peaks obscuring their quantification.”

Lines 204-209: “It should be pointed out that in our previous study on R. stricta alkaloids twelve major alkaloids, from twenty hairy root clones, were quantified by GC-MS analyses [23]. However, in the current investigation ten alkaloids were quantified by GC-MS. The two remaining alkaloids, rhazine and yohimbine isomer I, showed considerable peak broadening and partial overlapping with the neighbouring peaks, therefore, their quantification were remained obscure and did not included in the analyses.”

Question 3:

Third, just signaling changes in profiles of organic, amino, and fatty acids upon elicitation is, in my opinion, insufficient. Since the authors do have access to GC-MS, obtaining profiles of these compounds in control and elicited cultures should be relatively easy, and providing a more detailed description of the changes will definitely increase the value of this manuscript.

Answer 3:

We agree that a broader analytical coverage would be interesting but quantification of all compounds in the respective classes would each require dedicated methods and go far beyond the scope of this article. Here the focus was primarily on quantifiable effects on alkaloids. However, we mention the elicitation effects on primary metabolites to stimulate further work specifically dedicated to these phenomena.

Question 4:

On the related note, the quality of the supplied figures is quite poor - they look somewhat hazy; maybe this is because of down sampling for the peer review copy, but the authors should look into this.

Answer 4:

The figures are supplied in Tiff-format and should be fine in the printed version.

Question 5:

Also, the contents of TIAs are in the less than ng to a few ng/mg DW range, but the axes of graphs are in μg/mg. Get rid of these zeros, please.

Answer 5:

A very relevant comment. The axes are now changed to ng/mg in order to add clarity.

We are confident that the manuscript has been significantly improved and hope that it is now considered for publication in “Plants”.

With kind regards

Reviewer 2 Report

At least in my humble opinion, this manuscript has a great potential to be extremely well cited (in terms of its hetero-citations), once when published.

Last but not least, very best of (research) luck ahead to You all.

Author Response

Reviewer 2:

Comments and Suggestions for Authors

Question 1:

At least in my humble opinion, this manuscript has a great potential to be extremely well cited (in terms of its hetero-citations), once when published. Last but not least, very best of (research) luck ahead to you all.

Answer 1:

We appreciate the positive attitude. Thank you very much.

Reviewer 3 Report

The manuscript entitled "Methyljasmonate elicitation increases terpenoid indole alkaloid accumulation in Rhazya stricta hairy root cultures" provides an interesting results regarding the biosynthesis of indole alkaloids in the root of the medicinal plants Rhazya stricta. I have some few comments regarding this study:

1-it would be nice to see the change in the molecular level of these Indole alkaloid, thus authors my try to examine the gene expression of some genes in the indole alkaloid pathway under meJA treatment and control plants.

2-Its better to show the chromatogram and MS signals for major compounds in the supp file.

3-PCA figure, its better to assign the total variance on each principal components not as shown in the figure.

Author Response

Dear Gill Tan,

We appreciate the opportunity to revise our manuscript according to your and the reviewer’s valuable criticism. Please find a point-by-point list of answers addressing all requests below. Additionally, the English of the manuscript has been carefully edited (cf. track changes) to fulfill the high standards of “Plants”.

Reviewer 3:

Comments and Suggestions for Authors

Question 1:

The manuscript entitled "Methyljasmonate elicitation increases terpenoid indole alkaloid accumulation in Rhazya stricta hairy root cultures" provides an interesting results regarding the biosynthesis of indole alkaloids in the root of the medicinal plants Rhazya stricta. I have some few comments regarding this study:

1-it would be nice to see the change in the molecular level of these Indole alkaloid, thus authors my try to examine the gene expression of some genes in the indole alkaloid pathway under meJA treatment and control plants.

Answer 1:

We agree that transcriptomics could be interesting to follow up on the presented observations. From experience, we know that such experiments, incl. combination of several -omics, are extensive in non-model plants such as Rhazia and warrant own dedicated projects. Our current investigation provides the basis for such work and we hope that such activities will be inspired by our manuscript even though the biosynthetic pathways to the alkaloids are far from being completely resolved and only few involved genes are known.

Question 2:

2-It is better to show the chromatogram and MS signals for major compounds in the supp file.

Answer 2:

We have included the requested information as supporting information.

Question 3:

3-PCA figure, its better to assign the total variance on each principal components not as shown in the figure.

Answer 3:

We indeed prefer to present the data in Fig. 3 as it is now to illustrate the best way the effect of MeJA elicitation to our hairy roots. The same visualization pattern has been used in the following articles, too:

1 H-NMR metabolite fingerprinting analysis reveals a disease biomarker and a field treatment response in Xylella fastidiosa pauca-infected olive trees. Plants 2019, 8, 115. Effect of benzothiadiazole on the metabolome of tomato plants infected by citrus exocortis viroid. Viruses 2019, 11, 437. Identification of antiplasmodial triterpenes from Keetia species using NMR-based metabolic profiling. 2019, 15, 27. Metabolic changes of salicylic acid-elicited Catharanthus roseus cell suspension cultures monitored by NMR-based metabolomics. Biotechnol Lett. 2009, 31, 1967–1974. Elicitation studies in cell suspension cultures of Cannabis sativa J. Biotechnol. 2009, 143, 157–168. Quality Assessment of Ginseng by 1H NMR Metabolite Fingerprinting and Profiling Analysis. Agric. Food Chem. 2009, 57, 7513–7522. Quantitative 1H NMR metabolomics reveals extensive metabolic reprogramming of primary and secondary metabolism in elicitor-treated opium poppy cell cultures. BMC Plant Biol. 2008, 8, 1–19. Metabolomic alterations in elicitor treated Silybum marianum suspension cultures monitored by nuclear magnetic resonance spectroscopy. J Biotechnol. 2007, 130, 133–142.

We are confident that the manuscript has been significantly improved and hope that it is now considered for publication in “Plants”.

With kind regards

Round 2

Reviewer 1 Report

The authors corrected almost all the issues pointed out in my previous review. I particularly appreciate a more in-depth discussion of the effects of MeJa elicitation on the biosynthesis of TIAs as well as the contents of the new supplementary materials. In my opinion, the manuscript is significantly improved and can be considered as ready for publication.